# Loop Amplitudes from Precision Networks

Simon Badger[1], Anja Butter[2,3], Michel Luchmann[2], Sebastian Pitz[2] and Tilman Plehn[2]

**1** Physics Department, Torino University and INFN Torino, Torino, Italy
**2** Institut für Theoretische Physik, Universität Heidelberg, Germany
**3** LPNHE, Sorbonne Université, Université Paris Cité, CNRS/IN2P3, Paris, France

July 1, 2022

## Abstract

Evaluating loop amplitudes is a time-consuming part of LHC event generation. For di-photon production with jets we show that simple, Bayesian networks can learn such amplitudes and model their uncertainties reliably. A boosted training of the Bayesian network further improves the uncertainty estimate and the network precision in critical phase space regions. In general, boosted network training of Bayesian networks allows us to move between fit-like and interpolation-like regimes of network training.

# 1 Introduction

Combining our expectations of a vastly increased dataset from the upcoming LHC runs with novel analysis methods and ever-improving theory predictions, we are looking at exciting times for particle physics. One of the keys to make optimal use of the LHC data is to consistently employ modern techniques, inspired by data science and further developed for particle physics application. Inference based on precision predictions from first principles critically rests on the assumption that we can provide theory predictions over the full phase space fast, precisely, and with flexible model assumptions. To meet the speed and precision expectations from HL-LHC we can use modern machine learning (ML) throughout the event generation and simulation chain [1].

A straightforward ML-task is regression of loop amplitudes, represented as a smooth scalar function over a relatively simple phase space. For simple $(2 \to 2)$-processes learning a non-divergent loop amplitude does not even require deep networks [2] and has been achieved with conventional interpolation methods [3,4] as well. For higher final-state multiplicities [5] precision turns into a challenge, which we can try to meet by separating phase space into finite and divergent regions [6] or physics-inspired channels combined with very large training samples [7]. For our di-photon benchmark process at one-loop order current methods have shown to work well, but with limited precision especially in challenging regions of phase space [8].

Like in many physics applications, we would like to complement precision predictions of amplitudes with a reliable uncertainty estimate. Amplitudes are a simple problem because the training data consists of arbitrarily precise numerical values for well-defined phase space points. Once the network has learned all relevant features, we expect the leading uncertainties to reflect a possible local sparsity of the training data. Bayesian networks are perfectly suited to track training-related uncertainties [9]. In LHC physics they have been applied to regression [10], classification [11], ensembling [12], and generation [13,14]. We will use Bayesian networks as a surrogate for ML-amplitudes because they learn amplitude values together with an uncertainty, and because we can use their likelihood loss to improve the network training.

When we want to train a network on amplitudes over phase space, with the additional condition that large amplitude values should be reproduced well, we need to re-think our training strategy. While usual NN-applications can be viewed as a non-parametric fit, we want to precisely reproduce individual amplitudes in the spirit of an interpolation [15]. We can force the network to reproduce certain amplitudes by boosting these amplitudes in the training. To decide which amplitudes need boosting, we use a Bayesian network with its point-wise uncertainty estimate. We find that moving freely between fit-like and the interpolation-like tasks allows us to improve the uncertainty estimate through a loss-based boosting and the precision though a process-specific performance boosting.

In Sec. 2 we introduce our dataset and the benchmark results, before introducing the Bayesian network in Sec. 3. The improved training through the two boosting strategies is illustrated for the $\gamma\gamma g$ channel in Sec. 4. Finally, we compare a set of 1-dimension kinematic distributions for the training data and the NN-amplitudes including uncertainties and with the different boosting strategies, before we provide an Outlook. In the Appendix we show the corresponding results for the $\gamma\gamma gg$ final state, based on the same concepts and architectures, but using a larger network and more training data.

## 2 Dataset and benchmark results

As an example process for our surrogate NN-amplitudes we use the partonic one-loop process [8]

$$gg \to \gamma\gamma g(g) \,, \tag{1}$$

generated with SHERPA [16] and the NJET amplitude library [17]. We apply a basic set of detector-inspired cuts on the partons in the final state,

$$
\begin{aligned}
p_{T,j} &> 20 \text{ GeV} & |\eta_j| &< 5 & R_{jj,j\gamma,\gamma\gamma} &> 0.4 \\
p_{T,\gamma} &> 40, 30 \text{ GeV} & |\eta_\gamma| &< 2.37 \,.
\end{aligned} \tag{2}
$$

These cuts reduce the originally produced dataset from 1.1M $(2 \to 3)$-amplitudes of Ref. [8] to roughly 960k amplitudes. Each data point consists of real amplitude values as a function of the external 4-momenta, defining a 20-dimensional phase space. We divide our dataset into 90k training amplitudes and 870k independent test amplitudes acting as a high-statistics truth.

Transition amplitudes play a special role in the applications of neural networks to LHC physics, because they can be computed as functions of phase space with essentially arbitrary precision. The combination of high precision with limited training data is the challenge for the corresponding regression networks. Implicitly, it is assumed that the NN-amplitude networks will be faster than even the evaluation of leading-order amplitudes with state-of-the-art methods. The main figure of merit compares the true and the NN-amplitudes for a set of training or test data points,

$$\Delta_j^{\text{(train)}} = \frac{A_{j,\text{NN}}}{A_{j,\text{train}}} - 1 \qquad \text{or} \qquad \Delta_j^{\text{(test)}} = \frac{A_{j,\text{NN}}}{A_{j,\text{test}}} - 1 \,, \tag{3}$$

where $j$ runs over amplitude data points and we subtract 1 compared with the original paper [8]. Typical distributions of these $\Delta$ for existing calculations come with a width of 10% or more for the one-jet process of Eq.(25) [8]. For the tree-level process $e^+e^- \to q\bar{q}g$ the width of the $\Delta$-distribution can be reduced to the per-mille level, using a training dataset of 60M amplitudes and a rather complex, physics-inspired architecture of networks [7].

Our approach follows a different strategy from the physics-inspired architectures mentioned above. We will use a relatively simple and small network, enhanced by a Bayesian network structure, and target the precision requirements with a new training strategy. The goal is to show that small and hence fast networks are expressive enough to describe a scattering amplitude over phase space. Enforcing and controlling the required precision leads us to, essentially, an appropriate loss function and the corresponding network training strategy.

## 3 Bayesian network

Deterministic networks, usually trained by minimizing an MSE loss function, exhibit several weaknesses when it comes to LHC applications and controlled precision predictions. First, they only learn the amplitude value over phase space, without any information on if they have learned all features and how precise their estimate is. Second, their conceptually weak MSE loss function limits their performance. We will show how a Bayesian network with a likelihood comes with a whole range of conceptual and practical benefits.

---

In the main body of the paper we work with the $(2 \to 3)$-process, the corresponding results for the 4-body final state are presented in the Appendix. The interface between NJET and SHERPA is provided with Ref. [8] and available at https://github.com/JosephPB/n3jet.

**Bayesian networks and uncertainties**

In contrast to standard, deterministic networks, Bayesian neural networks (BNNs) learn distributions of network parameters or weights $\omega$ [9,18]. Sampling over the weight distributions gives us an uncertainty in the network output. At the end of this introduction we will approximate each weight distribution by a Gaussian, which does not limit the expressivity of a deep Bayesian network, but means that the Bayesian network requires only twice as many parameters as its deterministic counterpart [9]. By definition, the Bayesian network includes a generalized dropout and an explicit regularization term in the loss, which stabilize the training.

With our amplitude network we want to predict the transition amplitude $A$ for a phase space point $x$. If we define $p(A|x) \equiv p(A)$ as the probability distribution for possible amplitudes at a given phase space point $x$, and omitting the argument $x$ from now on, its mean value is

$$\langle A \rangle = \int dA\, A\, p(A) \qquad \text{with} \qquad p(A) = \int d\omega\, p(A|\omega)\, p(\omega|T)\,, \tag{4}$$

where $p(\omega|T)$ are the network weight distribution and $T$ is the training data. We do not know the closed form of $p(\omega|T)$, but we can approximate it with a simpler tractable distribution $q(\omega)$:

$$p(A) = \int d\omega\, p(A|\omega)\, p(\omega|T) \approx \int d\omega\, p(A|\omega)\, q(\omega)\,. \tag{5}$$

This approximation leads us directly to the BNN loss function. We implement the variational approximation as a Kullback-Leibler divergence,

$$\begin{aligned}
\text{KL}[q(\omega), p(\omega|T)] &= \int d\omega\, q(\omega)\, \log\frac{q(\omega)}{p(\omega|T)} \\
&= \int d\omega\, q(\omega)\, \log\frac{q(\omega)p(T)}{p(\omega)p(T|\omega)} \\
&= \text{KL}[q(\omega), p(\omega)] - \int d\omega\, q(\omega)\, \log p(T|\omega) + \log p(T) \int d\omega\, q(\omega)\,. \tag{6}
\end{aligned}$$

Bayes' theorem gives the corresponding networks their name. The prior $p(\omega)$ describes the model parameters before training. The model evidence $p(T)$ guarantees the correct normalization of $p(\omega|T)$. Turning Eq.(6) into a loss function we can omit the evidence, if we enforce

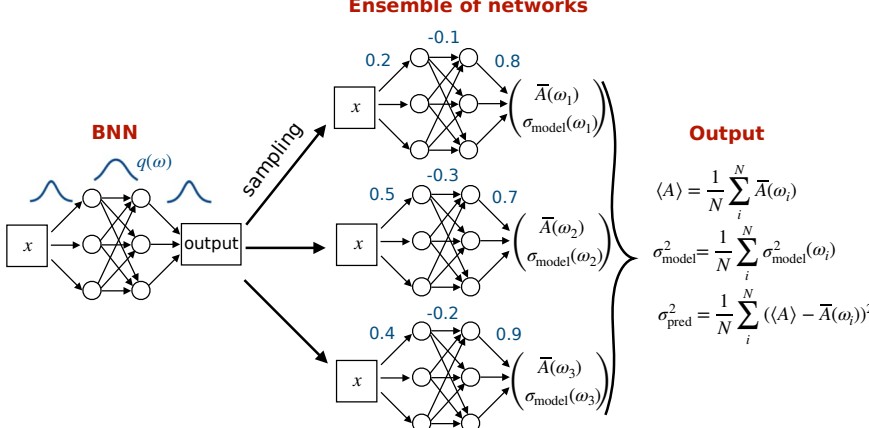

Figure 1: Illustration of the Bayesian network.

the normalization condition another way,

$$\mathcal{L}_{\text{BNN}} = -\int d\omega\, q(\omega)\, \log p(T|\omega) + \text{KL}[q(\omega), p(\omega)]\,. \tag{7}$$

The combined log-likelihood $\log p(T|\omega)$ implicitly includes the sum over all training points.

To get access to the mean and the uncertainty of the network prediction for $A$, we exchange the two integrals in Eq.(4) and find

$$\langle A \rangle = \int dA\, d\omega\, A\, p(A|\omega, T)\, q(\omega)$$

$$\equiv \int d\omega\, q(\omega)\, \overline{A}(\omega) \qquad \text{with} \qquad \overline{A}(\omega) = \int dA\, A\, p(A|\omega)\,. \tag{8}$$

A network with perfect $x$-resolution and perfect interpolation properties would be described by $q(\omega) = \delta(\omega - \omega_0)$, and $p(A|\omega)$ would simply return the one correct value for the amplitude. For noisy training data $p(A|\omega)$ actually describes a spectrum reflecting the noisy labels [10]. In our case, the amplitudes are exact, but the network will still not interpolate perfectly between the sparse training data points. Corresponding to Eq.(8) the variance of $A$ is

$$\sigma_{\text{tot}}^2 = \int dA\, d\omega\, (A - \langle A \rangle)^2\, p(A|\omega)\, q(\omega)$$

$$= \int d\omega\, q(\omega) \left[ \int dA\, A^2\, p(A|\omega) - 2\langle A \rangle \int dA\, A\, p(A|\omega) + \langle A \rangle^2 \int dA\, p(A|\omega) \right]$$

$$= \int d\omega\, q(\omega) \left[ \overline{A^2}(\omega) - 2\langle A \rangle \overline{A}(\omega) + \langle A \rangle^2 \right]$$

$$= \int d\omega\, q(\omega) \left[ \overline{A^2}(\omega) - \overline{A}(\omega)^2 + \left( \overline{A}(\omega) - \langle A \rangle \right)^2 \right] \equiv \sigma_{\text{model}}^2 + \sigma_{\text{pred}}^2\,, \tag{9}$$

where $\overline{A^2}(\omega)$ is defined in analogy to $\overline{A}(\omega)$ in Eq.(8). This defines two contributions to the uncertainty. First, $\sigma_{\text{pred}}$ is defined in terms of the $\omega$-integrated expectation value $\langle A \rangle$

$$\sigma_{\text{pred}}^2 = \int d\omega\, q(\omega) \left[ \overline{A}(\omega) - \langle A \rangle \right]^2\,. \tag{10}$$

It vanishes in the limit of perfect training (with infinite training data), $q(\omega) \to \delta(\omega - \omega_0)$. In that sense it represents a statistical uncertainty, and for a well-trained network we expect it to become small. In contrast, $\sigma_{\text{model}}$ already occurs without sampling the network parameters,

$$\sigma_{\text{model}}^2 \equiv \langle \sigma_{\text{model}}(\omega)^2 \rangle = \int d\omega\, q(\omega)\, \sigma_{\text{model}}(\omega)^2$$

$$= \int d\omega\, q(\omega) \left[ \overline{A^2}(\omega) - \overline{A}(\omega)^2 \right]\,. \tag{11}$$

It will be induced by limited training data, but in the limit of perfect training it approaches a plateau, accounting for a non-deterministic or stochastic label, limited expressivity of the network, not-so-smart choices of hyperparameters etc, in the sense of a systematic uncertainty. To avoid mis-understanding we refer to it as a model-related uncertainty rather than the usual $\sigma_{\text{stoch}}$ in case our data is non-stochastic, like the amplitudes in this application.

To understand these two uncertainty measures better, we can read Eqs.(8) and (11) as a sampling of $\overline{A}(\omega)$ and $\sigma_{\text{model}}(\omega)^2$ over a Gaussian network parameter distribution $q(\omega)$. This

sampling uses the network-encoded amplitude and uncertainty values over phase space $x$ and network parameter space $\omega$,

$$\text{BNN} : x, \omega \;\rightarrow\; \begin{pmatrix} \overline{A}(\omega) \\ \sigma_{\text{model}}(\omega) \end{pmatrix} . \tag{12}$$

Until now, we have not made any simplifying assumptions about the prior or weight distributions. To start with, in Ref. [11] we have shown that the details of the prior $p(\omega)$ have no visible effect on the network output. If we assume a Gaussian prior and, in addition, a Gaussian weight distribution $q(\omega)$ with the respective means and widths, the regularization term in Eq.(7) turns into

$$\text{KL}[q_{\mu,\sigma}(\omega), p_{\mu,\sigma}(\omega)] = \frac{\sigma_q^2 - \sigma_p^2 + (\mu_q - \mu_p)^2}{2\sigma_p^2} + \log \frac{\sigma_p}{\sigma_q} . \tag{13}$$

For this form we can use the reparameterization trick to translate an $\omega$-dependence into a dependence on $\sigma_q$ and $\mu_q$. Second, we can simplify the loss function by assuming that the $\omega$-dependent network output in Eq.(12) is described by a Gaussian. This allows us to approximate the likelihood $p(T|\omega)$ in Eq.(7) as Gaussian, and the BNN loss function becomes

$$\mathcal{L}_{\text{BNN}} = \int d\omega \, q_{\mu,\sigma}(\omega) \sum_{\text{points } j} \left[ \frac{\left| \overline{A}_j(\omega) - A_j^{(\text{truth})} \right|^2}{2\sigma_{\text{model},j}(\omega)^2} + \log \sigma_{\text{model},j}(\omega) \right]$$
$$+ \frac{\sigma_q^2 - \sigma_p^2 + (\mu_q - \mu_p)^2}{2\sigma_p^2} + \log \frac{\sigma_p}{\sigma_q} . \tag{14}$$

This loss is minimized with respect to the means and standard deviations of the network weights describing $q_{\mu,\sigma}(\omega)$. In this setup, the log-likelihood term includes a trainable uncertainty $\sigma_{\text{model}}(\omega)$ which is learned by the network in parallel to the amplitudes. When we evaluate the likelihood over a mini-batch rather than the full training dataset, we rescale the normalization of the regularization term to account for the different numbers of data points.

The same heteroscedastic loss [19] can be used in deterministic networks, if we introduce $\sigma_{\text{model}}$ as a second trained quantity in addition to the amplitude values. The Bayesian network setup guarantees that we really capture all training-related uncertainties correctly, at the expense of splitting the uncertainty measures $\sigma_{\text{model}}$ and $\sigma_{\text{pred}}$. It also does not make assumptions about a Gaussian uncertainty of the network output, so we stick to the more general BNN, even though it might well be possible to use a deterministic network for similar applications.

**Network architecture**

We use one Bayesian network trained on the entire training dataset. We train on amplitudes as a function of phase space with logarithmic preprocessing,

$$A_j \;\rightarrow\; \log\left(1 + \frac{A_j}{\sigma_A}\right) , \tag{15}$$

where $\sigma_A$ is given by the distribution of the amplitude values. In addition, all phase space directions are preprocessed by subtracting the respective mean and dividing by the respective standard deviation.

The network describing the $(2 \rightarrow 3)$-part of the reference process in Eq.(25) consists of four hidden layers with 20 kinematic input dimensions, $\{20, 20, 30, 40\}$ nodes, and two output dimensions corresponding to the amplitude and its uncertainty, as illustrated in Eq.(12). The network has around 6k parameters. Between the hidden layers we use a tanh activation function, while for the last layer we find that a SoftPlus activations outperforms GeLU slightly and ReLU significantly. The network is trained on 90k amplitudes for 400000 epochs with a batch size of 8192 and learning rate of $10^{-4}$, after which we observe no significant improvement in the loss. We use the Adam optimizer [20] with standard parameters.

**BNN performance**

As a first test of our BNN, we check the precision with which it approximates the true amplitudes in the training and test datasets, as defined in Eq.(3). For Fig. 2 we split the amplitudes by their absolute values, to see the effect of the limited training statistics in the collinear phase space regions. For the complete set of amplitudes the precision follows an approximate Gaussian with a width of a few per-mille, for the training and for the test data. This matches the best available performance from the literature [7], but with a very compact and fast network.

In the logarithmic panels of Fig. 2 we see that the tails of the $\Delta$-distributions for the full datasets are clearly enhanced. The picture changes when we only consider the phase space points with large amplitudes. For the 0.1% largest amplitudes the network is consistently less accurate with a slight tendency of underestimating the amplitudes. This is the motivation for training a separate network on the divergent phase space region(s) [8]. As we will see, the BNN offers an alternative approach which allows the full amplitude to be accurately described

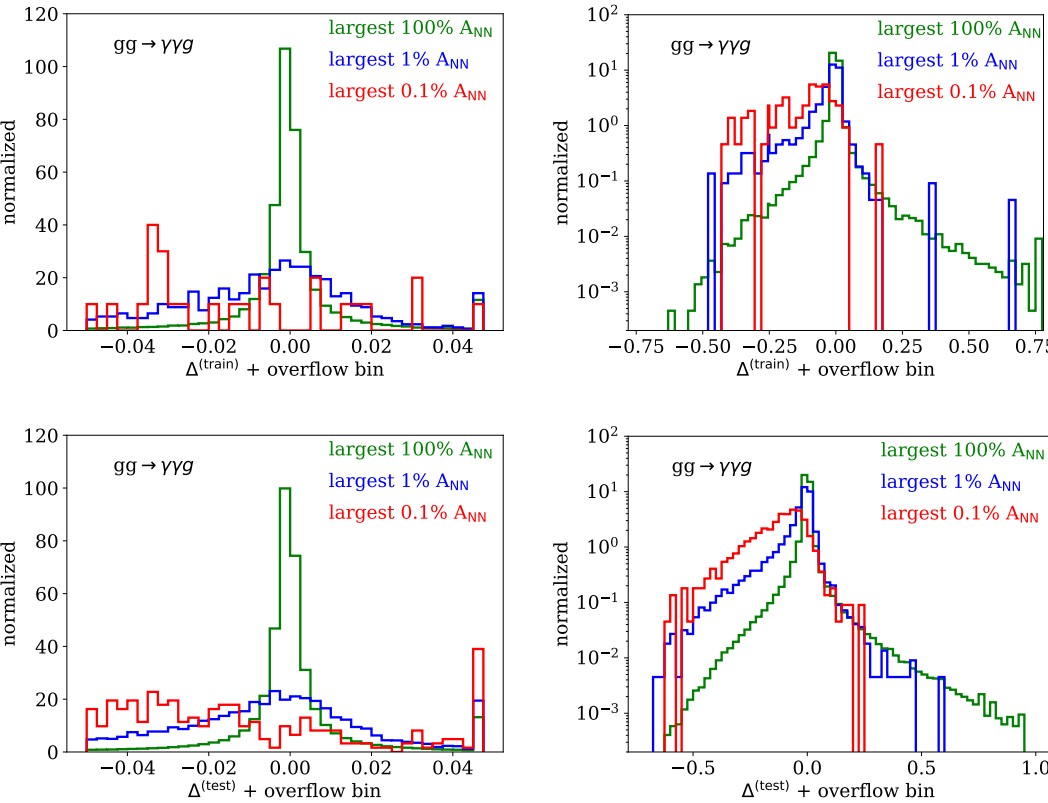

Figure 2: Performance of the BNN in terms of the precision of the generated amplitudes, Eq.(3), evaluated on the training (upper) and test datasets (lower).

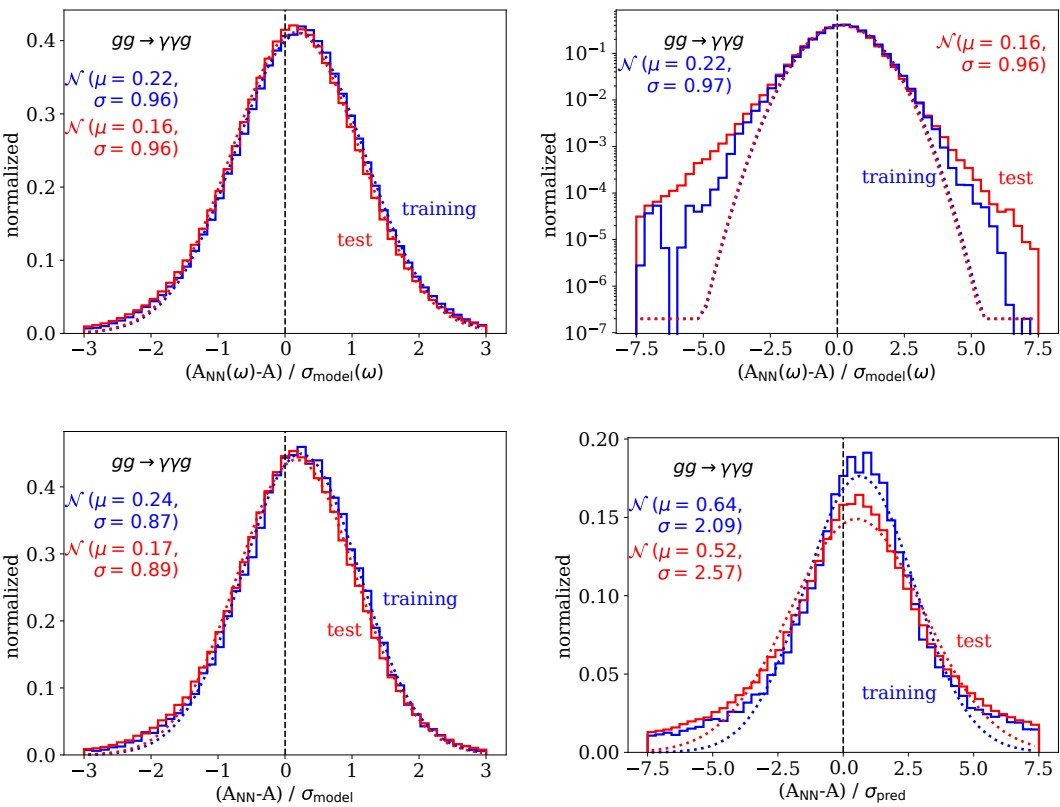

Figure 3: Pulls for the BNN, defined in Eq.(17) and evaluated on the training and test data. The two upper panels show the same curve for the weight-dependent pull on a linear and a logarithmic axis.

by a single network.

Because the BNN provides us with an uncertainty estimate for the NN-amplitude, we can define pull variables after integrating over the weight distributions,

$$t_{\text{model},j} = \frac{\langle A \rangle_j - A_j^{(\text{truth})}}{\sigma_{\text{model},j}} \qquad \text{or} \qquad t_{\text{pred},j} = \frac{\langle A \rangle_j - A_j^{(\text{truth})}}{\sigma_{\text{pred},j}} \,, \tag{16}$$

where the point-wise 'truth' refers to the training or test datasets we use to evaluate the pulls. Neither of these pulls have an $\omega$-dependent counterpart, because their numerators and denominators are sampled over the network weights independently. In the upper panels of Fig. 3 we see that the two pulls follow an approximate Gaussian shape, but with a much broader distribution for the $\sigma_{\text{pred}}$-based pull because of the smaller estimated uncertainty. We note that because of the log-rescaling of Eq.(15) it is not actually the amplitudes $A$ which should define Gaussian pulls, but their logarithms. We have explicitly checked that indeed the $\log A$ lead to a Gaussian, but that given our limited range of relevant amplitudes, the Gaussian shape translates into an approximately Gaussian shape for the amplitudes themselves.

Making use of the Gaussian likelihood loss of the BNN, Eq.(14), we can also define the weight-dependent pull

$$t_{\text{model},j}(\omega) = \frac{\overline{A}_j(\omega) - A_j^{(\text{truth})}}{\sigma_{\text{model},j}(\omega)} \,. \tag{17}$$

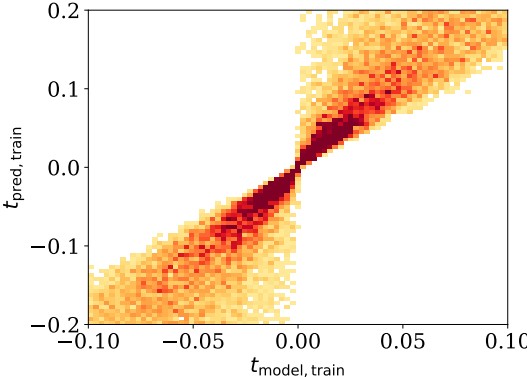

Figure 4: Correlation between the two pulls for the BNN, evaluated on the training data.

As part of the loss we can use its distribution as a consistency condition during network training. Given the Gaussian likelihood loss we expect a Gaussian distribution of $t_{\mathrm{model},j}(\omega)$, sampled over $\omega$ according to the Gaussian $q(\omega)$ and over phase space points $x$. In the upper panels of Fig. 3 we see that, again, the pull distribution is Gaussian in the center, but develops symmetric, enhanced tails roughly two standard deviations from the mean.

Finally, we need to go back to the definition of the network uncertainties and understand how the split $\sigma_{\mathrm{tot}}^2 = \sigma_{\mathrm{model}}^2 + \sigma_{\mathrm{pred}}^2$ can affect improved ways of training the amplitude network. We show the two sampled pulls defined in Eq.(16) in the lower panels of Fig. 3. Both are approximately Gaussian, and the width of the $t_{\mathrm{model},j}$ distribution is much smaller than for $t_{\mathrm{pred},j}$. This is an effect of the general observation that for a well-trained model

$$\sigma_{\mathrm{tot}} \approx \sigma_{\mathrm{model}} > \sigma_{\mathrm{pred}} \,. \tag{18}$$

The only issue with all pulls shown in Fig. 3 is that they come with a slight bias towards positive values, which means the network slightly overestimates the amplitudes as a whole. This is in contrast to the underestimation of the 0.1% largest amplitudes observed in Fig. 2, and we will target it by improving the network training.

Figure 4 shows very strong correlations between the two pulls defined in Eq. (16). Both pulls correctly identify the training data points which are not described by the network well. For our regression tasks with exact amplitudes both uncertainties are largely induced by the lack of training statistics especially in the divergent phase space regions, so this correlation is expected.

## 4 Network boosting

While the BNN-amplitude results described in the previous section are promising, the distribution of amplitudes and the pull distributions indicate potential improvements. We know that for generative networks we can employ an additional discriminator network to identify poorly learned phase space regimes [14], the solution is much simpler for a regression network. In the BNN loss we can compute the relative deviations between data and network output, or large pulls, and target these amplitudes directly. Once we control the network and its uncertainties, we can even think about further enhancing the training in the direction of an interpolation.

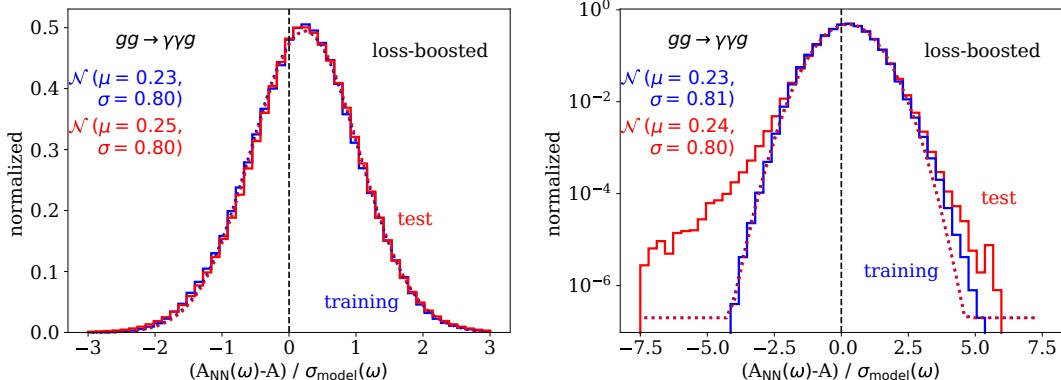

Figure 5: Pulls for the loss-boosted BNN, defined in Eq.(17) and evaluated on the training and test data. The two panels show the same results on a linear and a logarithmic axis. All curves can be compared to the BNN results without boosting in Fig. 3.

## 4.1 Loss-based boosting

Because the BNN loss in Eq.(14) represents a Gaussian log-likelihood, we can modify it and require a higher precision for those phase space points which according to the BNN uncertainty are not yet learned well. In practice, this is equivalent to feeding these training data points $n_j$ times into the computation of the BNN loss

$$\mathcal{L}_{\text{Boosted BNN}} = \int d\omega \, q_{\mu,\sigma}(\omega) \sum_{\text{points } j} n_j \times \left[ \frac{\left| \bar{A}_j(\omega) - A_j^{(\text{truth})} \right|^2}{2\sigma_{\text{model},j}(\omega)^2} + \log \sigma_{\text{model},j}(\omega) \right]$$
$$+ \frac{\sigma_q^2 - \sigma_p^2 + (\mu_q - \mu_p)^2}{2\sigma_p^2} + \log \frac{\sigma_p}{\sigma_q} \,. \tag{19}$$

As mentioned for Eq.(14), the regularization has to be adjusted for the additional amplitudes in the boosted training sample. This feedback training is similar to simple boosting algorithms for decision trees, where amplitudes for which the decision tree gives a wrong result are enhanced with additional weights. In our simple approach we duplicate some training amplitudes, or equivalently increases their weights in discrete steps.

In a first stage, we improve the self-consistency of the network with the initial assumptions and boost the network training for amplitudes with a large $\omega$-dependent pull, Eq.(17). In five iterations we identify the amplitudes with values of $t_{\text{model},j}(\omega)$, which are more than two standard deviations away from the mean and increase their contribution to the loss function in Eq.(19) by values $n_j = 5$. This is done four times. After adding the weights we continue the training on the enlarged datasets. For the next training cycle we again add weights to the amplitudes which now have pulls more than two standard deviations away. Each training ends when we see no more significant change to the loss which usually takes around 2000 epochs. This boosting forces the network towards a more self-consistent description of the tails of the pull distributions. We checked that small variations of $n_j$ or the number of cycles do not have a significant impact on these improvements.

In Fig. 5 we show the pulls from the boosted Bayesian neural network, boosted based on the self-consistency of the loss measured by the pull. We see a significant improvement for $t_{\text{model}}(\omega)$, the parameter we target with our boosting. One would naively expect the corresponding distribution to assume a Gaussian shape with unit width. However, first of all our

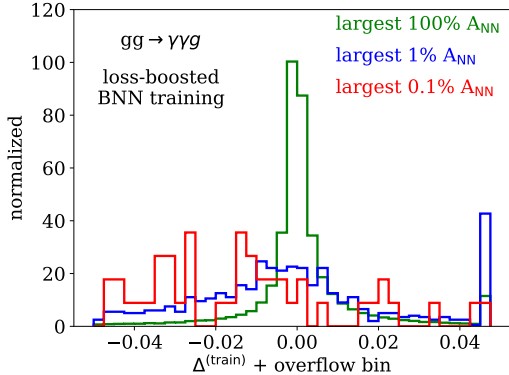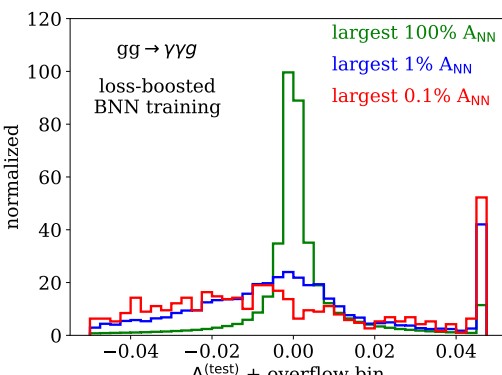

Figure 6: Performance of the loss-boosted BNN in terms of the precision of the gener-
ated amplitudes, Eq.(3), evaluated on the training and test datasets on a linear (left)
and logarithmic (right) axis. The curves can be compared to the BNN results without
boosting in Fig. 2.

loss-based boosting only moves amplitudes from the tails into the bulk, which means that the
tails of the boosted pull distributions should be low. Second, the pulls entering the loss and the
pulls shown in Fig. 5 are different because the loss includes weights for high-pull amplitudes.
In combination, both effects explain the narrower Gaussian for $t_{model}(\omega)$. In the logarithmic
version we also see a visible over-training though loss-boosting.

Moving on to the precision for the amplitudes, we see in Fig. 6 that the loss-boosting only
has a mild impact on the $\Delta$-distributions. It does not significantly improve the precision of the
amplitudes compared to Fig. 2, so we need a second boosting step.

## 4.2 Performance boosting

Given that the loss-boosting in the previous section worked for the uncertainty estimate but
only had a modest effect on the performance of our amplitude network, we proceed to a more
powerful boosting strategy. Independent of the self-consistency of the network, we know at
the training level which amplitudes challenge the network. This means we can select them

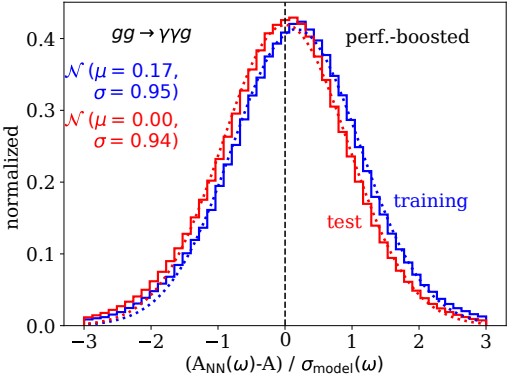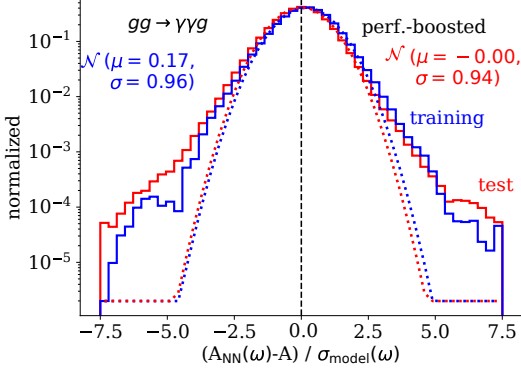

Figure 7: Pulls for the performance-boosted BNN, defined in Eq.(17) and evaluated
on the training and test data. The two panels show the same results on a linear and
a logarithmic axis. All curves can be compared to the BNN results without boosting
in Fig. 3 and the loss-boosted results in Fig. 5.

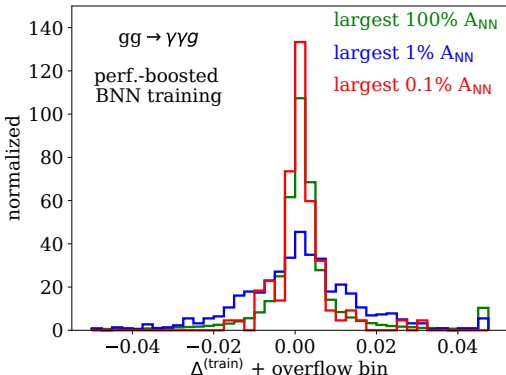 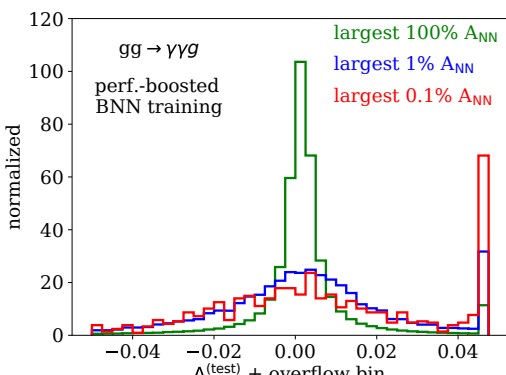

Figure 8: Performance of the performance-boosted BNN in terms of the precision of the generated amplitudes, Eq.(3), evaluated on the training (left) and test datasets (right). All curves can be compared to the BNN results without boosting in Fig. 2 and the loss-boosted results in Fig. 6.

with the goal of improving the training for the largest amplitudes. The difference between a general loss boosting and this process-dependent strategy is that now we target the largest and most poorly learned amplitudes by selecting them based on $\sigma_{\text{tot}}$. We choose the 200 amplitudes with the largest uncertainty $\sigma_{\text{tot}}$ and add three additional copies to the training dataset. This process is repeated 20 times, where each training ends when we see no more significant change to the loss which is usually around 2000 epochs.

In Fig. 7 we first see that the process-specific performance boosting broadens the pull distributions and this way reverses some of the beneficial effects of the loss-boosting on $t_{\text{model}}(\omega)$. However, the widths of the boost distributions remains below one, and the bias towards larger amplitudes is removed. This is true for the training data and for the test data. In addition, the consistency with the Gaussian shape is broken symmetrically for too small and too large amplitudes, again consistently for training and test data. Given that the two boostings target different amplitudes and effectively compete with each other, this pattern is expected.

The positive impact on the large amplitudes can be seen more clearly in Fig. 8. Evaluated on the training data, the 0.1% largest amplitudes now show a clear peak at small $\Delta^{\text{train}}$, consistent with all other amplitudes. This means the network has learned all amplitudes in the training dataset equally well. This effect translates to the test sample qualitatively, so the performance on the test data improves after performance-boosting, but this improvement is less pronounced than for the training data. This means that, at the expense of an overtraining, we have improved our network from a fit-like description to an interpolation-like description of the largest amplitudes.

The pattern observed by performance-boosting points to a conceptual weakness of standard network training when it comes to precision applications. If we stop the network training at the point where the performance on a training sample exceeds the performance on the test sample, we miss the opportunity of improving the network on the test and training data, but at a different rate. Overtraining is, per se, not a problem, as we know from applications of interpolation to describe data. The only challenge from such a network overtraining is a reliable uncertainty estimate from the generalization, for which we propose an appropriate scheduling of loss-boosting and performance-boosting.

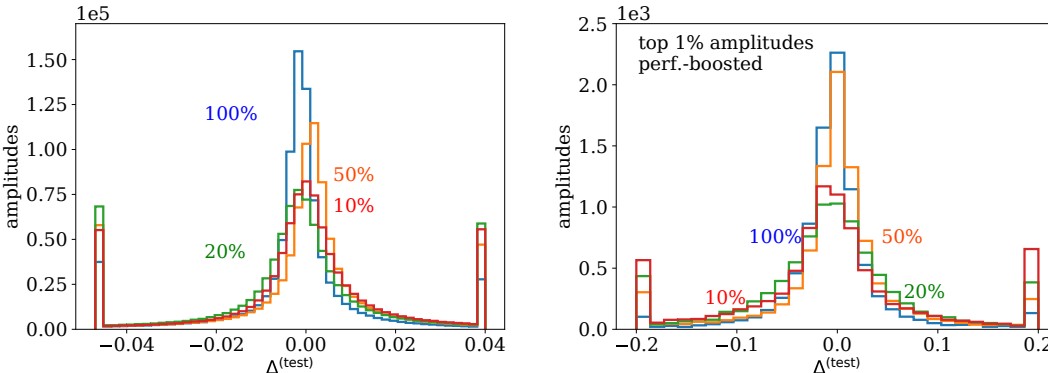

Figure 9: Performance of the BBN for all amplitudes (left) and a performance-boosted BNN for the largest 1% of all amplitudes (right), after training on different fractions of the full training dataset.

## 4.3 Effect of training statistics

Given that our amplitude-BNN has successfully learned the amplitudes for the partonic process $gg \to \gamma\gamma g$ well below the percent level, with a small and simple network and 90k training points, we can ask the question how much training data we actually need for a precision amplitude network. For this study we use the same BNN as before, including loss-boosting and performance-boosting, but trained on a reduced dataset of

$$10\% \quad (9.000 \text{ amplitudes}) \quad \cdots \quad 100\% \quad (90.000 \text{ amplitudes}) . \tag{20}$$

In Fig. 9 we show the corresponding $\Delta$-distributions for the test dataset. Our smallest training dataset contains 9000 amplitudes, which turn out sufficient to train our network with its 6192 parameters. The corresponding network reproduces the test data well, albeit with sizeable overflow bins. Increasing the amount of training data improves the precision of the network, but relatively slowly. We observe the same level of improvement for all amplitudes and for the 1% largest amplitudes. For the latter we only show results after process boosting, without any boosting the quality of the low-statistics training is comparably poor.

## 5 Kinematic distributions

After illustrating the performance of the amplitude network in a somewhat abstract manner, we can also show 1-dimensional kinematic distributions. The integration of the remaining phase space dimension requires a little care, because we cannot just integrate the uncertainties together with the central values for the amplitudes.

For the central values we combine the amplitudes with phase space sampling. For example applying the simple RAMBO [21] algorithm we identify the phase space weights with $A$. A 1-dimensional distribution is generated through bins which collect the sum of the amplitudes in the remaining phase space directions. The histogram value for a bin $k$ is

$$h_k = \sum_{j=1}^{N} A_j . \tag{21}$$

To use the amplitudes predicted by the BNN we have to add the sampling over the weights $\omega$. By replacing the truth amplitudes with the NN-amplitudes we can target the uncertainties

from the modelling of the amplitudes through the BNN. In analogy to Eq.(8) and omitting the index $k$ for the histogram we first extract a central histogram value as

$$\langle h \rangle = \int d\omega \, q(\omega) \sum_j \overline{A}_i(\omega)$$

$$= \int d\omega \, q(\omega) \overline{h}(\omega) \qquad \text{with} \qquad \overline{h}(\omega) = \sum_j \overline{A}_j(\omega) \,. \tag{22}$$

Again in analogy to Eqs.(10) and (11) we define the absolute uncertainties on the bin entry as

$$\sigma_{h,\text{pred}}^2 = \int d\omega \, q(\omega) \left[ \overline{h}(\omega) - \langle h \rangle \right]^2$$

$$\sigma_{h,\text{model}}^2 = \int d\omega \, q(\omega) \left[ \overline{h^2}(\omega) - \overline{h}(\omega)^2 \right] \,. \tag{23}$$

The total uncertainty is again $\sigma_{h,\text{tot}}^2 = \sigma_{h,\text{model}}^2 + \sigma_{h,\text{pred}}^2$. We can simplify $\sigma_{h,\text{model}}$ further. In all of the above formulas $h$ is just a sum over amplitudes. If we assume that the corresponding $\sigma_{\text{model}}$ values are uncorrelated, we can relate $\sigma_{h,\text{model}}$ to $\sigma_{\text{model}}$ by exchanging the sum and the variance,

$$\sigma_{h,\text{model}}^2 = \int d\omega \, q(\omega) \text{Var}(\overline{h}(\omega))$$

$$= \int d\omega \, q(\omega) \text{Var}\left( \sum_j A_j(\omega) \right) = \int d\omega \, q(\omega) \sum_j \text{Var}\left( A_j(\omega) \right)$$

$$= \int d\omega \, q(\omega) \sum_j \sigma_{\text{model},j}^2(\omega) \equiv \left\langle \sum_j \sigma_{\text{model},j}^2(\omega) \right\rangle \,. \tag{24}$$

While we assume uncorrelated uncertainties for $\sigma_{\text{model}}$ we cannot do the same for $\sigma_{\text{pred}}$. To compute $\sigma_{\text{pred}}$ we first sample a set of weight configurations, which turns our BNN into an ensemble of neural networks, and then use each of these neural networks to compute the corresponding histogram value. Computing the standard deviation of these values gives us an estimate for $\sigma_{h,\text{pred}}$. By sampling from the weight distributions we change the neural network itself and all of its predictions. To assume that these changes are uncorrelated for different amplitudes seems not exceptionally well justified.

Based on this procedure we show BNN-amplitude results for a set of kinematic distributions in the upper panels of Fig. 10. We see the effect of limited training data towards the end of the different kinematic distributions, where the agreement between the NN-amplitudes and truth deteriorates. For our reference process this happens for $|\eta_g| \gtrsim 2.5$ or $|\eta_\gamma| \gtrsim 1.5$. Still, the BNN uncertainty estimate covers the deviation from the truth reliably.

In the lower panels of Fig. 10 we see that after performance-boosting the BNN predictions agree with the training data spectacularly well. This is the goal of the boosting and leads to the network learning all features in the training data extremely well. In the phase space regions where the regular BNN precision is limited by sparse and large training amplitudes, the improved agreement between NN-amplitudes and the training data carries over to the test data at a level that the network prediction is significantly improved. The uncertainties for the training data still cover the deviations from the truth, but unlike the central values this uncertainty estimate does not generalize correctly to the test data. This structural issue with process boosting could be ameliorated by alternating between loss-boosting and performance-boosting, until the specific requirements of a given analysis in precision and uncertainty estimates are met.

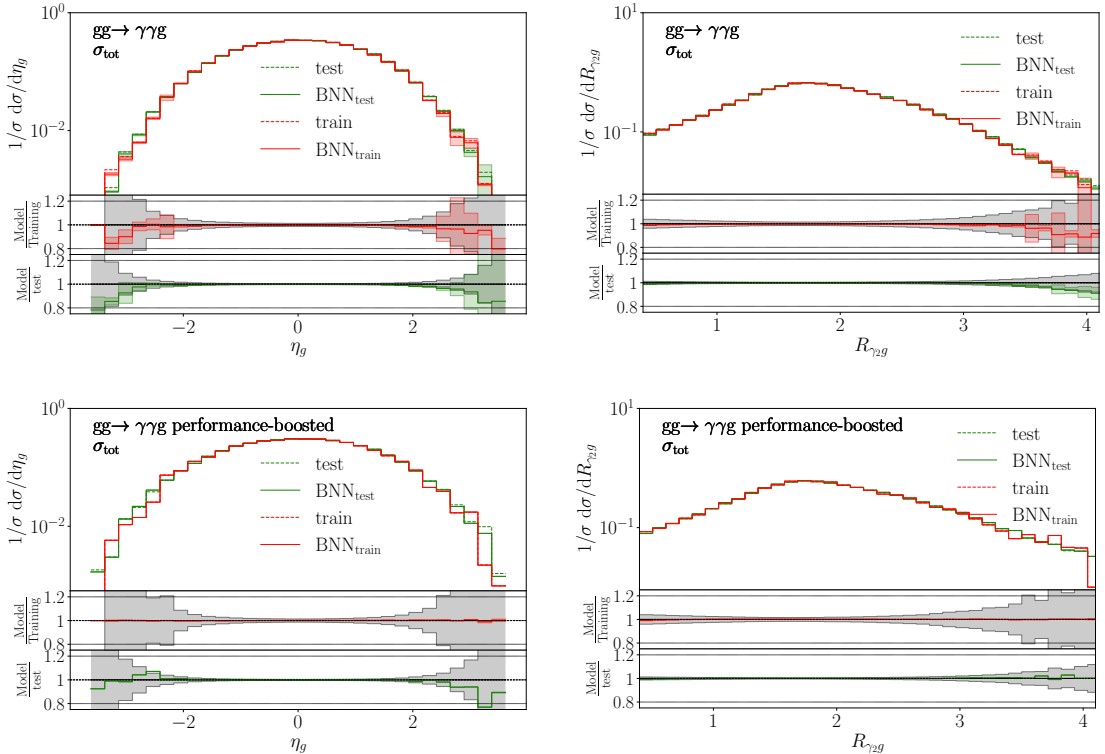

Figure 10: Kinematic distributions from the BNN without boosting (upper) and after performance-boosting (lower). The grey error bars in the lower panels indicate the statistical limitation of the training and test data.

## 6 Outlook

Learning loop-amplitudes for LHC simulations is a classic ML-problem, because we need to train a precision network only once to provide a much faster simulation tool which can be used many times. In this application neural networks really work like better fits to the training data. Unlike for many other network applications, the training amplitudes are not noisy, which means we would like to reproduce the training amplitudes exactly, supplemented with a controlled uncertainty over all of phase space. To provide a reliable uncertainty map over phase space, we can rely on Bayesian regression networks [10].

The precision task reminds us of an interpolation rather than a fit, which means we need modify our ML-approach conceptually. If we are willing to accept a certain amount of overtraining, we can significantly improve the network training through boosting certain amplitudes. Because the Bayesian network provides a reliable uncertainty estimate, we can select the to-be-boosted amplitudes based on their deviation from the training data in units of the uncertainty. This loss-based boosting simply improves the self-consistency of the Bayesian network training. In a second step, we can boost training amplitudes just based on their absolute uncertainty. This selection helps with the precision for a given process, and because we use the absolute uncertainty we typically focus on the largest amplitude values.

We have applied Bayesian network training and the two strategies of amplitude boosting to the partonic process $gg \to \gamma\gamma g$ [8]. We have first found that the network amplitudes agree with the true amplitudes at the sub-percent level, for the training data and for a test dataset. For the 1% largest amplitudes an agreement at the percent level required process-specific performance boosting. For 1-dimensional kinematic distributions we have seen that

the performance-boosting allows for extremely precise predictions in kinematic tails, albeit with a somewhat reduced performance in the uncertainty estimate for the test dataset. This can be improved by alternating between process and loss boosting in order to retain improved uncertainty estimation and increased performance which will be subject of future studies.

Finally, we have checked what happens with our boosted Bayesian network training when we reduce the number of training amplitudes from 90k to 9k and found that thanks to the boosteing this only leads to a mild decrease in the network precision. This leaves us confident that boosted amplitude training with its shift from a fit-like to interpolation-like objective provides us with highly efficient surrogate models whenever the generation of training data is CPU-intensive.

## Acknowledgments

First, we would like to thank Manuel Haussmann for introducing us to Bayesian networks. We thank Steffen Schumann for many enlightening discussions and Frank Krauss for regularizing any unreasonable enthusiasm. We are also grateful to Joseph Aylett-Bullock and Ryan Moodie for helpful conversations. This research is supported by the Deutsche Forschungsgemeinschaft (DFG, German Research Foundation) under grant 396021762 – TRR 257: *Particle Physics Phenomenology after the Higgs Discovery*, through Germany's Excellence Strategy EXC 2181/1 - 390900948 (the Heidelberg STRUCTURES Excellence Cluster) and the European Union's Horizon 2020 research and innovation programmes *High precision multi-jet dynamics at the LHC* (consolidator grant agreement No 772009).

# A    $(2 \rightarrow 4)$-process

We know that that increasing the number of particles in the final state leads to a significant drop in network performance [6–8, 22, 23]. We illustrate how we can use the boosted BNN training to improve network predictions for the $(2 \rightarrow 4)$-process

$$gg \rightarrow \gamma\gamma gg \,. \tag{25}$$

For this process we use a network with seven hidden layers, 24 kinematic input dimensions, $\{32, 64, 256, 512, 128, 64, 32\}$ nodes, and two output dimensions corresponding to the amplitude and its uncertainty. This larger network has around 600k parameters. The training dataset contains around 90k amplitudes. Aside from these changes, we apply the same basic BNN training with two levels of loss boosting and process-specific performance boosting.

As for the $(2 \rightarrow 3)$-process we first show the performance of the network training in Fig. 11. While the overall scale of the agreement has increased for the sub-percent level to the percent level, we still see that the network has learned the largest amplitudes extremely well after process boosting. Unlike for the standard BNN, there is a certain amount of overtraining after performance-boosting, indicating the shift from a fit-like network training to an interpolation-like training.

Next, we check the consistency of the network output by looking at the $\omega$-dependent pull distribution defined Eq.(17). We see that especially for large amplitudes the loss-boosting guarantees a well-behaved, consistent network, while the additional process boosting reverses

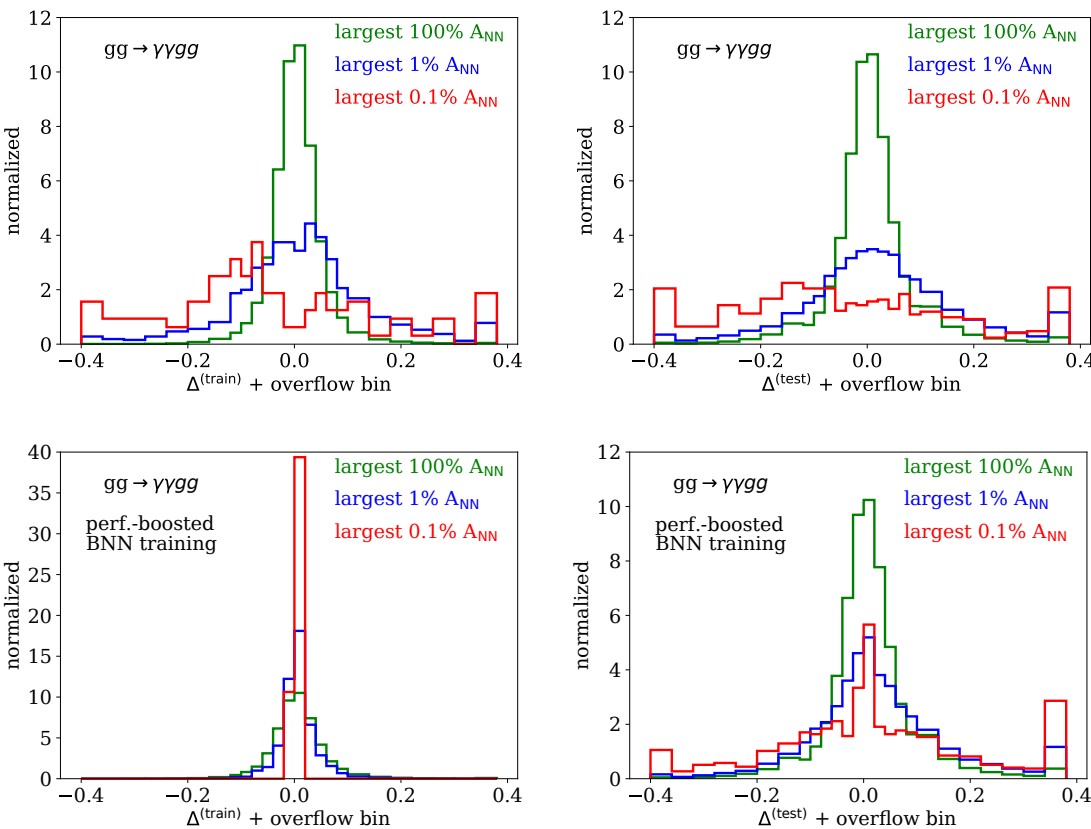

Figure 11: Performance of the basic (upper) and performance-boosted (lower) BNNs for the $(2 \rightarrow 4)$-process in terms of the precision of the generated amplitudes, Eq.(3), evaluated on the training (left) and test datasets (right).

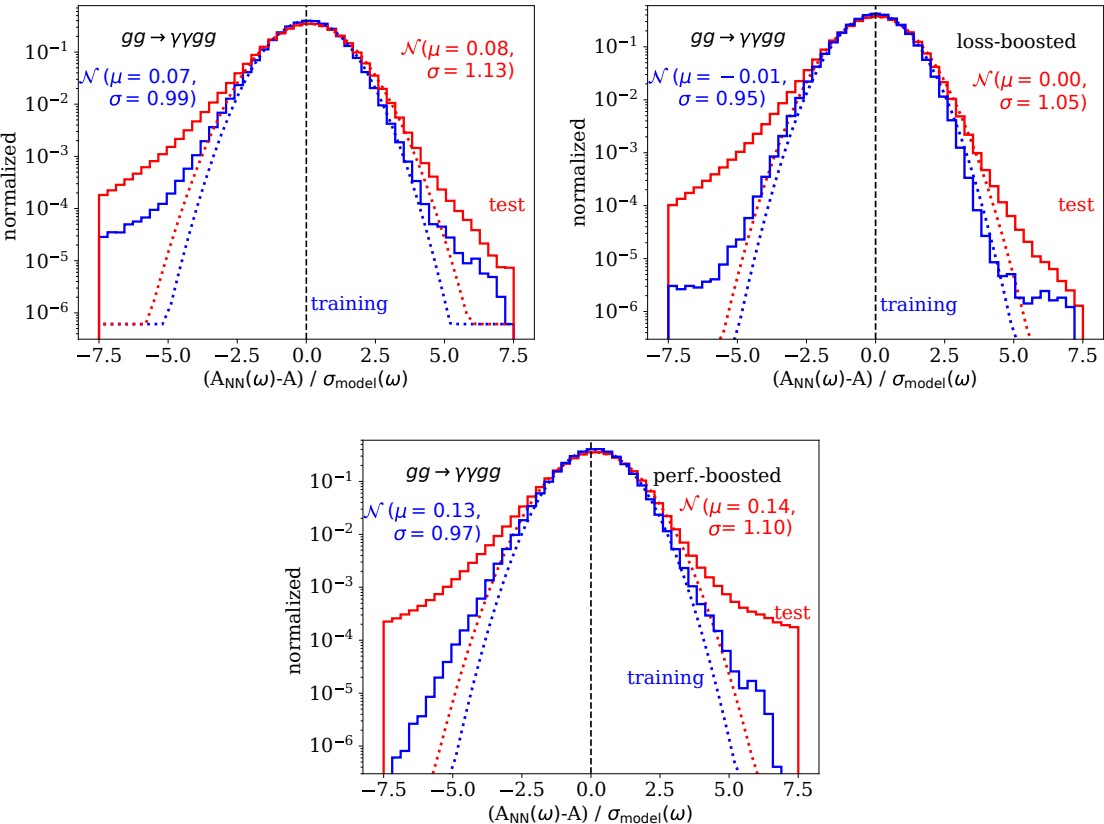

Figure 12: Pulls of the loss-bossted (left), and performance-boosted (right) BNN for the $(2 \rightarrow 4)$-process, defined in Eq.(17) and evaluated on the training and test data.

some of the beneficial effects of the loss-boosting. This effect was already observed for the $(2 \rightarrow 3)$-process.

Finally, we show the 1-dimensional kinematic distributions for the basic BNN and for the performance-boosted BNN. As for the $(2 \rightarrow 3)$-process the boosting step has a spectacula effect on the training data in the poorly learned kinematic tails. After integrating over the additional phase space directions this improvement translates well into the test dataset, but at the expense of the uncertainty estimate on the training data.

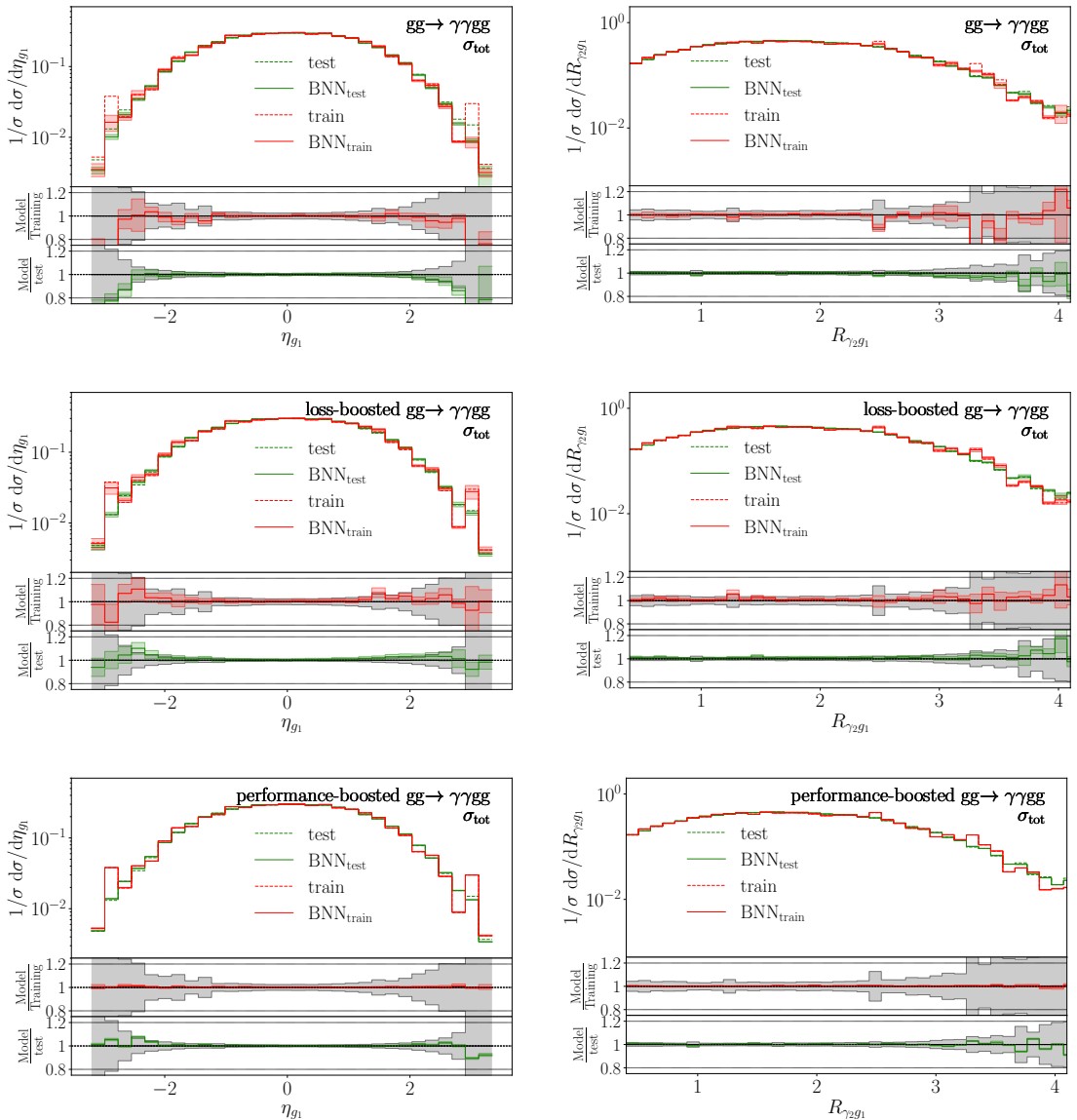

Figure 13: Kinematic distribution for the $(2 \to 4)$-process without boosting (upper), after loss boosting (center), and after process boosting (lower). The grey error bars in the lower panels indicate the statistical limitation of the training and test data.

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
