# Peer review of "Loop Amplitudes from Precision Networks"

_SciPost Physics Core_

## Round 1 · Referee Report · Anonymous (Referee 1) · 2022-9-10

Report

This paper describes the application of Bayesian neural networks to the
modelling of physical amplitudes. As a result, the architecture is able to
simultaneously model the amplitude, and an aleatoric and epistemic uncertainty.
Furthermore, the uncertainties are used to define boosting strategies that are
shown to significantly improve network performance.

I believe the presented methods and results certainly warrant publishing, as
they make an important contribution to the objective of pushing machine learning
applications in particle physics further from the proof-of-principle stage, and
towards practically-usable implementations. High degrees of precision and
control over uncertainties are vital for this objective. However, among some
other concerns, the explanation of Bayesian networks requires some improvement.

Requested changes

1- In section 2, the authors describe the dataset they use, which is taken from [8]. A set of cuts is then applied, but no justification for them is presented. I understand that one must make a choice in the application of cuts to avoid singularities. However, since later on the paper shows that the performance deteriorates for large amplitude, i.e. presumably close to the cut boundaries, one might be led to believe that the cuts are selected to enhance performance. It would be useful if the BNN could for instance be trained on looser or tighter cuts, to find out of the qualitative conclusions would be different.

As a side-question, why are the amplitudes real rather than complex numbers?

2- Why do the authors choose to make use of a 20-dimensional
representation of the phase space, while its actual dimension is only
7-dimensional (2 initial-state momentum fractions, and $3 \times 3 - 4 = 5$
final-state components)? Would a lower-dimensional representation not
improve performance?

3- At the beginning of section 3, it is claimed that the MSE loss limits
the performance of implementations without Bayesian architectures. What is
the justification for this claim? In particular, the BNN loss also has a MSE
term that is responsible for incentivizing the BNN to match predicted and
real amplitudes.

4- Under \textbf{Bayesian networks and uncertainties}, there is a
sentence 'By definition, the Bayesian network includes a generalized dropout
and an explicit regularization term in the loss, which stabilize the
training. I believe the authors mean, respectively, the sampling over the
network weights and the simultaneous output of $\sigma_{\text{model}}$, but
this is certainly not clear to the reader at this point in the text, and may
still be unclear after reading the full paper if one does not have the
prerequisite domain knowledge.

5- Further down, there is a sentence 'We implement the variational
approximation as a KL divergence.' This sentence has similar issues of
requiring the reader to be familiar with the lingo. I think the explanation
should state that $p(\omega|T)$ is the real distribution that is unknown, so
you introduce a trainable variational approximation $q(w)$, and define a
loss function that minimizes the difference.

6- Before eq. 7, it is stated that the evidence can be dropped 'if the
normalization condition is enforced another way'. I do not believe that this
'other way' is clarified elsewhere. I believe that it should just say that
the evidence term does not depend on any trainable parameters, and can thus
be dropped from the loss.

7- While the authors already attempt to do so, I think the distinction
between $\sigma_{\text{model}}$ and $\sigma_{\text{pred}}$ could be
clarified further. My understanding is that

8- The next paragraph then makes a link with the actual implementation of
the BNN. However, it says 'This sampling uses the network-encoded amplitude
and uncertainty values...', while the reader does not even know yet that the
model is meant to predict those. I would reorder some elements of this
section, by first saying that the BNN is essentially meant to model $p(A|x,
\omega)$, $\bar{A}$ and $\sigma_{\text{model}}$ are its mean and variance,
and you can thus implement it as a feed-forward network that predicts that
mean and variance as a function of $x, \omega$.

9- Under \textbf{Network architecture}, is the uncertainty enforced to be
positive after the final $40 \to 2$ linear layer through some activation?
Does the mean have an activation?

10- Figures 2 and 3, it took me a while to figure out what the differences
were between the plots. Maybe in figure 2 add some text like '(train)' and
'(test)' underneath the $gg \to \gamma \gamma g$ labels. In figure 3 it is
especially confusing that the weight-dependent pull is shown above the
weight-independent pull, while the reverse is true in the text. I would
suggest splitting it into two figures, and adding some text to indicate
which pull is being plotted (even though it is also shown on the x-axis).

11- The results before boosting show a bias towards positive values. It is
unclear to me if this is also captured by the model uncertainty. Please
elaborate.

12- I think the reference to [14] should be supplemented by other works
that use this technique, like 2009.03796, 2106.00792. The comma after [14]
should also be a period.

13- Above section 4.2, 'over-training though loss-boosting' $\to$
'over-training through loss-boosting'

---

## Round 1 · Referee Report · Anonymous (Referee 2) · 2022-9-28

Report

The manuscript describes how Bayesian networks can be used to learn and reproduce loop amplitudes relevant for high-energy particle scattering. The trained network can provide both an estimate of the amplitude and an uncertainty estimate. The authors describe how a "boosting" procedure can be used to enhance the performance of the network and demonstrate their method on a well-known, but challenging, process.

I believe that the work presented in the manuscript is significant and addresses an important question. The methods presented could in principle be applied to accelerate the evaluation of the matrix elements entering, for example, simulations required for the LHC and future colliders. Indeed, the results displayed in Figure 10 are very impressive. In my opinion, this work is certainly suitable for publication in SciPost Core.

Below, I attach a few questions and comments that I would encourage the authors to address prior to the publication of this paper.

Requested changes

1 - Section 2, what quarks run in the loop? Are top/bottom quarks included and, if so, are the masses fixed or left as free parameters of the network?

2 - Throughout the draft the authors appear to use the word "amplitudes" to mean "samples of the amplitude"/"amplitude samples". In typical usage, one might say that the gg -> gam gam g process has a single amplitude (or perhaps several helicity or color-ordered amplitudes). However, the authors refer to "90k training amplitudes". Here I suspect they mean a single (or few) amplitudes sampled at 90k phase-space points.

3 - Section 2, the authors write that each data point consists of a real amplitude. Usually, an amplitude is a complex number. I suspect the authors are referring to the squared amplitude. If this is correct, this should be stated more clearly.

4 - The authors use a highly-redundant 20-dimensional parametrization. Why do the authors not use, for example, just the independent Mandelstam invariants? Can the authors demonstrate that a lower dimensional parametrization is not better for their training (as one might naively expect)?

5 - Below Eq(6) the authors write "if we enforce the normalization condition another way" in order to justify dropping the final term in Eq(6). The exact procedure is not clear to me, where is the normalization condition enforced? Perhaps this sentence can be reworded for clarity.

6 - In the first line of Eq(8) the authors refer to p(A|w,T). Perhaps I am misunderstanding or missing a step, is this a typo or does this indeed follow from Eq(4)?

7 - In the second paragraph of "Network architecture" the authors write "The 2->3 part of the reference process in Eq(25)", do they mean to reference instead Eq(1)?

8 - Comparing the \delta^(test) panel of Figure 2 with that of Figure 6 it naively appears that significantly more test points fall into the overflow bin (for the largest 0.1% and 1% of amplitude values) after loss-based boosting. Could the authors please comment further on this and to what extent do they consider it a problem?

9 - In Figure 8, although the \delta^(test) performance does seem to be broadly improved, again, significantly more test points fall into the overflow bin than in Figure 2 or Figure 6. Could the authors comment further on this?

10 - Sec 5, final paragraph, the authors write that "The uncertainties for the training data still cover the deviations from the truth, but unlike the central values this uncertainty estimate does not generalize correctly to the test data". If I have understood Figure 10 correctly, this fact is visible in the bottom (model/test) plot of the lower panel, where the green band no longer reflects the true uncertainty (which is presumably ~the grey band). One of the strengths of the authors' approach is that it provides not only an estimate of the amplitude but also of the uncertainty of the estimate. The authors write "This structural issue with process boosting could be ameliorated by alternating between loss-boosting and performance-boosting", can the authors demonstrate that an additional loss-boosting step would improve the quality of the uncertainty estimate without reversing the improvement in the performance? This would be a very strong and convincing argument for using their proposed procedure.

11 - Sec 6, there is a typo "boosteing"

---

## Editorial Decision

resubmitted